# Phospholipase A1 Member A Activates Fibroblast-like Synoviocytes through the Autotaxin-Lysophosphatidic Acid Receptor Axis

**DOI:** 10.3390/ijms222312685

**Published:** 2021-11-24

**Authors:** Yang Zhao, Stephan Hasse, Myriam Vaillancourt, Chenqi Zhao, Lynn Davis, Eric Boilard, Paul Fortin, John Di Battista, Patrice E. Poubelle, Sylvain G. Bourgoin

**Affiliations:** 1Centre de Recherche du CHU de Québec, Université Laval, Québec, QC G1V 4G2, Canada; yang.zhao@crchudequebec.ulaval.ca (Y.Z.); stephan.hasse@crchudequebec.ulaval.ca (S.H.); myriam.vaillancourt@crchudequebec.ulaval.ca (M.V.); chenqi.zhao@cervo.ulaval.ca (C.Z.); lynn@joanisse.net (L.D.); eric.boilard@crchudequebec.ulaval.ca (E.B.); paul.fortin@crchudequebec.ulaval.ca (P.F.); patrice.poubelle@crhudequebec.ulaval.ca (P.E.P.); 2Centre ARThrite, Université Laval, Québec, QC G1V 4G2, Canada; 3Département de Microbiologie-Infectiologie et d’Immunologie, Université Laval, Québec, QC G1V 0A6, Canada; 4Département de Médecine, Université Laval, Québec, QC G1V 0A6, Canada; 5Department of Medicine, Faculty of Medicine and Health Sciences, McGill University, Montréal, QC H3G 2M1, Canada; john.dibattista@mcgill.ca

**Keywords:** phosphatidylserine-specific phospholipase A1, lyso-phospholipid, lysophosphatidic acid, lyso-phosphatidylserine, autotaxin, chemokines, inflammation, arthritis

## Abstract

Lysophosphatidylserine (lysoPS) is known to regulate immune cell functions. Phospholipase A1 member A (PLA1A) can generate this bioactive lipid through hydrolysis of sn-1 fatty acids on phosphatidylserine (PS). PLA1A has been associated with cancer metastasis, asthma, as well as acute coronary syndrome. However, the functions of PLA1A in the development of systemic autoimmune rheumatic diseases remain elusive. To investigate the possible implication of PLA1A during rheumatic diseases, we monitored PLA1A in synovial fluids from patients with rheumatoid arthritis and plasma of early-diagnosed arthritis (EA) patients and clinically stable systemic lupus erythematosus (SLE) patients. We used human primary fibroblast-like synoviocytes (FLSs) to evaluate the PLA1A-induced biological responses. Our results highlighted that the plasma concentrations of PLA1A in EA and SLE patients were elevated compared to healthy donors. High concentrations of PLA1A were also detected in synovial fluids from rheumatoid arthritis patients compared to those from osteoarthritis (OA) and gout patients. The origin of PLA1A in FLSs and the arthritic joints remained unknown, as healthy human primary FLSs does not express the PLA1A transcript. Besides, the addition of recombinant PLA1A stimulated cultured human primary FLSs to secrete IL-8. Preincubation with heparin, autotaxin (ATX) inhibitor HA130 or lysophosphatidic acid (LPA) receptor antagonist Ki16425 reduced PLA1A-induced-secretion of IL-8. Our data suggested that FLS-associated PLA1A cleaves membrane-exposed PS into lysoPS, which is subsequently converted to LPA by ATX. Since primary FLSs do not express any lysoPS receptors, the data suggested PLA1A-mediated pro-inflammatory responses through the ATX-LPA receptor signaling axis.

## 1. Introduction

Phospholipase A1 member A (PLA1A, also known as PS-PLA1) belongs to the pancreatic lipase family and was purified initially from the thrombin-activated platelet-rich plasma of rats [1]. PLA1A shows specific substrate preference for sn-1 fatty acids on phosphatidylserine (PS) or lysophosphatidylserine (lysoPS). This feature distinguishes PLA1A from other classical pancreatic lipase family members [1]. PS localization in the inner leaflet of cell membrane lipid bilayer prevents this lipid from hydrolysis by secreted phospholipases, including PLA1A. However, when cells undergo activation or apoptosis, PS is exposed on the outer leaflet, presenting this specific lipid substrate to extracellular PLA1A to produce 2-acyl-lysoPS [2].

PLA1A is associated with several disease states [3]. For instance, high PLA1A expression was reported in the airway epithelium in asthma [4], in vascular endothelial cells of heart transplant rejection [5], as well as in non-metastasis melanoma cell lines [6] and prostate cancer tissues [7]. In this regard, the expression of PLA1A is not only restricted to inflammatory disorders [8] but also associated with oncogenesis and cancer metastasis [9]. Bronchoalveolar lavage fluids from certain allergic asthma patients [4], serum of melanoma subjects [10], Graves’ disease patients [11], and systemic lupus erythematosus (SLE) patients [8] contain elevated levels of PLA1A. Serum PLA1A significantly correlated with plasma lysoPS level in culprit coronary arteries from acute coronary syndrome patients [12,13] and in ascites from gastric cancer patients [14]. The production of lysoPS, which has immune-suppressive properties [14], may at least in part explain the biological functions of PLA1A. 

In humans, lysoPS mediates its effects through activation of its receptors, including GPR34 (LPS1), P2RY10 (LPS2), and GPR174 (LPS3) [15]. The best-characterized response to lysoPS is histamine release and eicosanoid production through stimulation of GPR34 in mast cells [16]. LysoPS also induces eosinophil degranulation by activating P2RY10 [17], promotes the clearance of apoptotic neutrophils by activating receptor G2A on macrophage [18], and also negatively influences regulatory T-cell accumulation through GPR174 [19]. During the development of atherosclerosis, lysoPS accelerates the formation of foam cells [20]. The presence of lysoPS on surface of platelets supports the assembly of prothrombinase complex [21]. Thus, lysoPS may play a role in regulating blood clotting, inflammation, and immune responses since hematopoietic cells highly express several lysoPS receptors.

In addition to its regulatory functions, lysoPS hydrolysis by autotaxin (ATX) is a potential source of extracellular lysophosphatidic acid (LPA). ATX is a member of nucleotide pyrophosphatase/phosphodiesterases first characterized as a cell motility factor [22]. In platelets, internalized ATX is released during cell activation and can bind to platelet αVβ3 and αIIbβ3 integrins [23]. Almost half of serum LPA is converted from lysophospholipids by ATX [24] in a platelet-dependent pathway [25]. LPA mediates its effects through activation of its six receptors LPA1–6. The ATX-LPA receptor axis has been extensively studied in vascular system development [26], in the regulation of lymphocyte homing [27], in cancer cell growth and metastasis [28,29,30], as well as in the development of cardiovascular diseases [31]. The sn-1 and the sn-2 positions of phospholipids are usually occupied by saturated and unsaturated fatty acids, respectively [32]. Therefore, the higher potency of unsaturated LPA species in mediating biological activities [33] reminds the importance of PLA1 isozymes responsible for unsaturated LPA species production [24].

Although PLA1A is likely responsible for the production of unsaturated lysoPS species, the contribution of PLA1A during the development of inflammatory diseases, such as rheumatoid arthritis (RA), remains largely unknown. In RA, the most characterized change is the inflammatory process occurring in the synovial tissue, in which fibroblast-like synoviocytes (FLSs) proliferate and invade contiguous cartilage [34]. Activation of FLSs contributes to the buildup of chronic synovial inflammation and cartilage destruction by producing various inflammatory mediators and matrix-degrading enzymes [34]. 

In the present study, we report on high concentrations of PLA1A in the plasma from EA and clinically stable SLE patients as well as in synovial fluids from RA patients. Normal human FLSs were used to objectively evaluate the pro-inflammatory effects of PLA1A. We monitored the expression of lysoPS receptors and used pharmacological inhibitors or antagonists of the ATX-LPA receptor axis to study the mechanisms by which PLA1A activates FLSs. We demonstrate that the pro-inflammatory effects of PLA1A in FLSs were, at least in part, dependent on the ATX-LPA receptor axis.

## 2. Results

### 2.1. PLA1A Levels in Plasma, Synovial Fluids, and Human Primary FLSs

We first examined by ELISA the concentrations of PLA1A in platelet-free plasma from 12 healthy donors, 38 early-diagnosed arthritis (EA) patients, and 62 SLE patients undergoing treatment as well as in hyaluronidase-treated synovial fluids from five RA patients, three psoriatic arthritis (PsoA) patients, three osteoarthritis (OA) patients, and three gout patients. PLA1A concentrations in plasma from EA patients were 4.46 ± 2.03 ng/mL (Figure 1A). Compared to healthy donors (0.09 ± 0.09 ng/mL) and clinically stable SLE patients (0.6 ± 0.2 ng/mL), EA patients showed a higher concentration of PLA1A in their plasma. There was no sex difference in the plasma concentration of PLA1A between male (3.44 ± 2.45 ng/mL) and female (5.49 ± 3.3 ng/mL) EA patients (Figure 1B). The concentrations of PLA1A in female SLE patients (0.67 ± 0.23 ng/mL) and males (0.19 ± 0.18 ng/mL) were not significantly different (Figure 1C). 

In synovial fluid samples, PLA1A was 35.08 ± 7.56 pg/mL in RA patients, 636.8 ± 521.4 pg/mL in PsoA patients, 10.04 ± 0.98 pg/mL in OA patients, and 18.74 ± 9.03 pg/mL in gout patients, respectively (Figure 1D). Compared to gout and the non-rheumatoid disease OA, the inflammatory synovial fluids of RA patients showed a higher concentration of PLA1A. The concentration of PLA1A in PsoA was higher than in RA synovial fluids.

Human tissues, such as skeletal muscle, kidney, liver, and testis, express the PLA1A transcript. However, the expression of PLA1A in human FLSs remains unknown. Higher PLA1A levels in synovial fluids of RA patients might come from FLSs. To assess this possibility, we performed a semi-quantitative RT-PCR to examine the expression of PLA1A in normal human primary FLSs using two different pairs of PLA1A oligo primers and the total RNA from human testis as a positive control. As shown in Figure 2A, PLA1A mRNA was not detectable in normal FLSs under our experimental conditions. However, using western blotting and flow cytometry, we detected PLA1A at the protein level in cultured FLSs (Figure 2B,C). In the whole-cell lysate, the amount of PLA1A in serum-starved cells was slightly decreased compared to serum-cultured cells, thereby suggesting that fetal bovine serum (FBS) is an extracellular source of PLA1A in cell culture. Flow cytometry without permeabilization failed to detect the PLA1A signal on the FLSs surface (data not shown). Flow cytometry with permeabilization detected the PLA1A signals in cells cultured with or without serum, and the amount of PLA1A was slightly lower in serum-starved cells.

### 2.2. Pro-Inflammatory Effects of PLA1A and the Detection of Its Substrate

To verify the potential pro-inflammatory effects of PLA1A, we incubated serum-starved cultured FLSs with or without recombinant PLA1A for 24 h (Figure 3). The basal concentration of IL-8 secretion in non-stimulated serum-starved FLSs was 4.03 ± 1.53 pg/mL. The addition of 0.2 µg/mL PLA1A increased the concentration of IL-8 to 17.76 ± 4.58 pg/mL. PLA1A induced IL-8 secretion in a concentration-dependent manner, and the highest concentration tested (0.5 µg/mL) had no cytotoxic effect as assessed using PI labeling. 

PLA1A specifically hydrolyzes PS, which usually localizes in the inner leaflet of the cell membrane. When cells are activated or undergoing apoptosis, they expose PS on the outer leaflet of the cell membrane. To assess the externalization of PS during cell activation, we treated FLSs with 100 ng/mL TNF for 1, 5, 15, 30, and 60 min and monitored PS exposure on FLSs by flow cytometry using FITC Annexin V. Cell viability was not affected under those conditions, with dead cells (PS- and PI-positive) representing less than 2% of the population. FITC Annexin V-positive and PI-negative cells were considered viable cells presenting PS on their outer leaflet. As shown in Figure 4A, the percentage of PS-positive FLSs increased from 13.98% to 23.6% (*p*-value > 0.05, n = 4), following stimulation with TNF for 30 min, and remained elevated up to 60 min, the last timepoint tested. 

PLA1A has an affinity to surface heparin sulfate proteoglycan, and added heparin can competitively bind to PLA1A, thereby preventing the cell-surface-exposed PS from hydrolysis [35]. Next, we added increasing amounts of heparin to FLSs 30 min before the treatment with or without 0.2 µg/mL PLA1A. As shown in Figure 4B, heparin concentrations less than 200 µg/mL enhanced PLA1A-mediated IL-8 production. At higher concentrations (400, 800, and 1600 µg/mL), heparin showed an inhibitory effect on IL-8 production. Since cells treated with heparin at a concentration higher than 800 µg/mL showed cell apoptosis (as observed through a microscope), we used heparin at 800 µg/mL in the subsequent experiments.

### 2.3. Pro-Inflammatory Effects of LysoPS and PLA1A in FLSs

PLA1A specifically hydrolyzed surface-exposed PS to produce lysoPS, which can bind and activate the lysoPS receptors in an autocrine or a paracrine manner. To determine whether the pro-inflammatory effects induced by PLA1A in cultured FLSs were dependent on the production of lysoPS and the activation of its receptors, we performed semi-quantitative RT-PCR to examine the expression of lysoPS receptors (GRP34, GPR174, and P2RY10) in human primary FLSs. As shown in Figure 5A, none of the lysoPS receptor transcripts was detected under our experimental conditions. In contrast, we detected the expression of GRP34, GPR174, and P2RY10 mRNAs in the human CD4^+^ T memory cells used as a positive control. 

Since there was no expression of lysoPS receptors in human FLSs, the data suggested that PLA1A-mediated IL-8 secretion was through another pathway. To investigate the mechanism underlying PLA1A-induced IL-8 production in healthy human FLSs, serum-starved cells were incubated with LPA (5 µM), lysoPS (5 µM), or PLA1A (0.2 µg/mL) in the absence or presence of LPA1/3 receptor antagonist Ki16425 (5 µM), ATX inhibitor HA130 (1 µM), or heparin (800 µg/mL), respectively. The concentration of lysophospholipids and other compounds was chosen based on preliminary results and as previously described [36]. LPA-, lysoPS-, and PLA1A-stimulated FLSs produced 18.17 ± 7.19, 55.65 ± 22.6, and 23.11 ± 7.44 pg/mL of IL-8, respectively. IL-8 secretion stimulated by LPA, lysoPS, and PLA1A were significantly decreased by Ki16425 by 90%, 65%, and 85%, respectively (Figure 5B–D). HA130 inhibited IL-8 secretion by 45% and 76% in cells stimulated with lysoPS and PLA1A, respectively (Figure 5C,D). We did not test concentrations of HA130 higher than 1 µM due to cell cytotoxicity or off-target effects (changes in cell morphology and chemokine synthesis). As expected, heparin attenuated PLA1A-induced IL-8 secretion by 67% (Figure 5D).

### 2.4. Effects of PLA1A and ATX in the Presence of Albumin

According to our data, the pro-inflammatory effects of PLA1A relied on the activity of ATX provided by FLSs. To define whether exogenous ATX promotes further cytokine production, we incubated the cultured FLSs with PLA1A (0.2 µg/mL) and recombinant ATX (5 nM) in combination. The basal level of IL-8 released by unstimulated FLSs cultured in media supplemented with or without 1% fatty acid-free BSA was 7.08 ± 2.45 and 8.05 ± 2.04 pg/mL, respectively. As illustrated in Figure 6A, in albumin-free condition, the addition of recombinant ATX alone had no effect on the production of IL-8, and PLA1A alone stimulated the production of IL-8 by approximately 2.5-fold. The combination of PLA1A and ATX increased the IL-8 production by 3.8-fold compared to non-stimulated cells, by 4.1-fold compared to ATX-treated cells, and by 1.5-fold compared to cells activated by PLA1A alone. However, differences were not statistically significant (*p*-value > 0.05). The addition to the culture media of 1% fatty acid-free BSA, which protects LPA from degradation and increases its half-life, enhanced the production of IL-8 in cultured FLSs. Compared to FLSs incubated in albumin-free media (Figure 6A), the presence of albumin (Figure 6B) increased ATX-induced IL-8 production by 2.3-fold, increased PLA1A-induced IL-8 secretion by 1.5-fold, and increased combination of ATX and PLA1A-induced IL-8 release by 1.8-fold.

## 3. Discussion

High PLA1A serum level and tissue expression are associated with cancer metastasis, acute coronary syndrome, and autoimmune diseases, such as SLE. The current study is the first to explore the pro-inflammatory effects of PLA1A in RA disease processes, using primary human FLSs. In detail, we show that (1) PLA1A was elevated in the plasma of EA patients and synovial fluids from RA patients; (2) even if PLA1A mRNA was not detected in FLSs, a high level of cell-associated PLA1A was detected in FLSs cultured in medium supplemented with FBS; (3) PLA1A hydrolyzed PS exposed on the outer leaflet of FLSs and enhanced secretion of IL-8 through a pathway independent of lysoPS receptors; (4) the pro-inflammatory effects of PLA1A and lysoPS on FLSs were induced through the ATX-LPA receptor (LPAR) axis; and (5) the combined effects of PLA1A and ATX on cytokine secretion were more than additive, and the addition of fatty acid-free BSA during cell stimulation further enhanced IL-8 production by FLSs. 

In our study, we first observed elevated concentrations of PLA1A in plasma of EA and synovial fluids from RA patients. PLA1A was increased in the circulation (49.5-fold increase in EA patients compared to healthy donors) and in the joint compartment (3.5-fold increase in RA patients compared to OA patients). There was no gender difference of plasma PLA1A concentration among EA patients. In clinically stable SLE patients, the plasma concentration of PLA1A was slightly higher in females. Differences in PLA1A concentration between active autoimmune disease and steady or non-autoimmune disease were also remarkable, as highlighted in a previous clinical study with SLE patients [8]. Our data showed that compared to clinically stable SLE patients, plasma PLA1A in EA patients was increased by 7.4-fold. These data suggested that the elevated concentrations of PLA1A might contribute to the inflammatory conditions in the EA patients. In this regard, PLA1A could be a promising biomarker to monitor inflammatory disease activity [8]. In the study by Sawada et al., there was no significant increase in PLA1A level in the serum of RA patients [8]. However, our study monitored PLA1A in EA patients (symptoms ≤ 12 months) who had not received any disease-modifying antirheumatic drugs. Furthermore, joint manifestations in patients with early disease are difficult to distinguish from other forms of inflammatory polyarthritis. It is not excluded that the early stage of the disease and uncontrolled inflammation could contribute to the high plasma PLA1A level in EA patients.

In healthy donors, we detected 0.09 ± 0.09 ng/mL of PLA1A in the platelet-free plasma, whereas 33.8 ± 16.6 µg/L of PLA1A in serum were measured using a newly developed enzyme immunoassay in a previous study [37]. The cellular source of plasma PLA1A is yet to be established. We cannot exclude a role for activated platelets as a source of plasma PLA1A in humans. Although PLA1A mRNA was not detectable in human platelets, the transcripts were present in skeletal muscle, kidney, small intestine, spleen, testis, and in several human cell lines [38]. Extracellular PLA1A could be associated with platelets or stored intracellularly, as reported for ATX [39]. In rat platelets, PLA1A stored in α-granules is secreted into the extracellular environment upon activation with thrombin [1,40]. In addition, the liver tissue might be a source of human serum PLA1A. The serum PLA1A level in patients with liver injury (hepatitis, cirrhosis) was increased in comparison to that of healthy subjects [41]. Furthermore, to show that liver tissue could be the source of serum PLA1A, Uranbileg et al. studied patients with hepatocellular carcinoma, a condition associated with higher levels of serum PLA1A in comparison to healthy subjects [41]. In those patients, serum levels of PLA1A were correlated with the expression of PLA1A mRNA in the liver tissue without any carcinoma (or background tissue) but were not correlated with the PLA1A mRNA expression in carcinoma tissues [41]. According to our RT-PCR results, primary human FLSs do not express the PLA1A mRNA. However, cultured FLSs may internalize the PLA1A present in culture medium supplemented with FBS. The PLA1A in synovial fluids may derive from plasma leakage associated with inflammation or from immune cells that infiltrate synovial tissues. 

PS has been widely involved in physiological and pathological conditions, such as blood coagulation, phagocytosis by macrophage, and activation of intracellular enzymes, such as protein kinase C [40]. Our results showed that TNF activation enhanced the exposure of PS in primary human FLS. This finding was consistent with previous reports suggesting that during platelet activation, cell apoptosis, and cytokine stimulation, PS externalized on the outer membrane leaflet is a potential substrate for secreted PLA1A [42]. Thus, PLA1A could hydrolyze externalized PS and contribute to IL-8 secretion through the ATX-LPA receptor axis (Figure 7). IL-8 is known to be a neutrophil chemotactic factor [43], and inhibition of IL-8 can reduce the neutrophil infiltration in RA joints [44]. The inhibition of IL-8 synthesis could also protect the cartilage and bone destruction and lead to disease suppression in collagen-induced arthritis model [45]. The N-terminal domain of PLA1A drives recognition and binding of the serine residue of PS and is responsible for its catalytic activity [38,40]. PLA1A belongs to the lipase family and has an affinity for heparin [46]. We should highlight that a low concentration of heparin enhanced PLA1A-mediated IL-8 secretion. Although the addition of heparin likely prevents PLA1A from binding to heparan sulfate lipid expressed by FLSs [47], we cannot exclude the possibility that PLA1A binding to heparan sulfate would modulate its catalytic activity. Our data highlighted that PLA1A had pro-inflammatory effects through hydrolyzing PS exposed on the surface of FLSs.

LysoPS receptors are expressed in immune cells. For instance, GPR34 is highly expressed in mast cells, whereas P2RY10 and GPR174 are expressed in lymphoid organs [15]. Our results showed that human FLSs expressed none of these three lysoPS receptors, thereby suggesting that lysoPS stimulated IL-8 secretion through an alternative pathway in FLSs and not through lysoPS receptor-induced signaling. ATX and LPA1/3 are widely expressed in human FLSs [36,48]. Indeed, our study showed significant inhibition of PLA1A- and lysoPS-mediated IL-8 production in the presence of an LPA1/3 antagonist (Ki16425) or an ATX inhibitor (HA130). Thereby, the data suggested that PLA1A and lysoPS mediated their pro-inflammatory effects through the ATX-LPAR axis. Thus, ATX metabolized extracellular PLA1A-derived lysoPS into LPA, which initiated a signaling cascade through activation of LPA1 and LPA3 expressed in FLSs [36].

We also found that ATX and PLA1A in combination enhanced IL-8 secretion in FLSs compared to cells stimulated with ATX or PLA1A alone. In albumin-free condition, the combination of ATX and PLA1A can increase the production of IL-8 to a certain extent but cannot largely increase the IL-8 production as in the albumin-containing condition. It is a clinically significant observation since both PLA1A and ATX are present in RA synovial fluids, and synovial fluid is an albumin-containing biofluid. Surprisingly, in an incubation media devoid of albumin, the addition of ATX had no significant effect on IL-8 secretion by FLSs. Previous studies reported that ATX-derived LPA could activate cells in an autocrine/paracrine manner [49]. On the one hand, cells release a significant proportion of newly synthesized ATX in a vesicle-bound form rather than a soluble form [50]. ATX binds to exosomes through its NUC domain and is catalytically active [50]. On the other hand, secreted-ATX can interact with cell integrins through its SMB domain [51], which localizes ATX on cell surfaces [50,52] and contributes to the synthesis of LPA in the vicinity to its cognate receptors [53]. Besides, LPA can repress the expression and activity of ATX, and LPA binding to delipidated albumin could not affect this inhibitory effect [54,55]. However, a high concentration of ATX substrate and inflammatory cytokines could attenuate the inhibitory feedback loop of LPA [55]. Our data suggest that ATX-derived LPA should be either transported to activate its receptors and/or should be protected from degradation by cell-surface lipid phosphate phosphatases [51]. Our results also illustrate the importance of albumin in the maintenance of LPA functional responses during cell activation. 

Unsaturated LPA species are produced by PLA1 isozymes and have been reported to be more potent than saturated LPA [33,46] in inducing proliferation [56], de-differentiation [42], and phenotypic modulation [57] of smooth muscle cells both in vivo and in vitro. ATX prefers lysophospholipids with unsaturated fatty acids according to the structure of the hydrophobic lipid-binding pocket [51,58]. Previous studies on the structure-activity relationship of cloned LPA receptors have reported differential activation of LPA receptors by different LPA species [33]. These differences include fatty acid length, degree of unsaturation, and linkage to the glycerol backbone. Cells and tissues express distinct patterns of LPA receptors. For example, LPA3 shows a higher affinity for C18:1, 18:2, and 18:3 LPA species compared to 20:4 LPA, whereas LPA1 shows a broad ligand specificity for either saturated or unsaturated LPA species [33]. These findings underline the potential importance of PLA1A as a source of unsaturated lysophospholipid species in various pathological conditions. Nevertheless, the origin of secreted PLA1A and the functions of PLA1A in healthy conditions and chronic rheumatic autoimmune diseases need further investigation.

## 4. Materials and Methods

### 4.1. Reagents 

1-Oleoyl-2-hydroxy-sn-glycerol-3-phosphate (LPA) and 1-oleoyl-2-hydroxy-sn-glycerol-3-phospho-L-serine (lysoPS) were from Avanti Polar Lipids (Alabaster, AL, USA). The lysophospholipid stocks were in phosphate-buffered saline supplemented with 0.1% fatty acid-free BSA and stored at −20 °C. Purified recombinant human phospholipase A1 (PLA1A) was from OriGene Technologies Inc. (Rockville, MD, USA). Recombinant autotaxin (ATX) and ATX inhibitor HA130 were from Echelon Biosciences Inc. (Salt Lake City, UT, USA). Human phospholipase A1 (PLA1) ELISA kit was from MyBioSource Inc. (San Diego, CA, USA). Human IL-8 ELISA Kit, PLA1A polyclonal antibody, Rabbit IgG isotype control were from Invitrogen Corporation (Frederick, MD, USA). Human TNF was from PeproTech Inc. (Rocky Hill, NJ, USA). Hyaluronidase from bovine testis, heparin sodium salt from porcine intestinal mucosa, and essential fatty acid-free bovine serum albumin (BSA) were from Sigma-Aldrich (St. Louis, MO, USA). LPA1/3 specific receptor antagonist Ki16425 was from Cayman Chemical (Ann Arbor, MI, USA). Antibody against the human GAPDH was from R&D Systems Inc. (Oakville, ON, Canada). Peroxidase-conjugated anti-rabbit or mouse IgG and PE-conjugated anti-rabbit IgG secondary antibodies were from Jackson ImmunoResearch Laboratories, Inc. (West Grove, PA, USA). Western Lightning chemiluminescence reagents were obtained from Perkin Elmer Life Sciences (Woodbridge, ON, Canada). Cell culture reagents and trypsin-EDTA cell detachment solution were purchased from Wisent Inc. (St-Bruno, QC, Canada). Propidium iodide (PI) and FITC Annexin V were from BD Biosciences (San Jose, CA, USA). Accutase cell detachment solution was from eBioscience (San Diego, CA, USA). Ribozol RNA extraction reagent from VWR International (Mississauga, ON, Canada). RNA to cDNA EcoDry Premix was obtained from Clontech Laboratories Inc. (Mountain View, CA, USA). SsoAdvanced Universal SYBR Green SuperMix was from Bio-Rad Laboratories (Mississauga, ON, Canada). Digitonin was from Abcam (Toronto, ON, Canada). 

### 4.2. Human Plasma and Synovial Fluid Samples

The CHU de Québec-Université Laval SARD Biobank and Data Repository provided the platelet-free plasma samples (12 healthy donors, 38 EA patients, and 62 undergoing treatment SLE patients). Patients with disease duration ≤ 12 months were considered EA patients. Human synovial fluid samples were obtained from five RA patients, three PsoA patients, three OA patients, and three gout patients. The CHU de Québec’s Ethics Committee approved the study. All samples were collected after informed consent and were stored at −80 °C until the measurements.

### 4.3. Cell Culture

Normal human primary FLSs were purchased from Asterand (Detroit, MI, USA) or provided by Dr. Di Battista (MUHC-Research Institute). Cells were maintained under standard conditions (37 °C and 5% CO_2_) and grown in DMEM supplemented with 10% FBS, penicillin (100 IU), and streptomycin (100 µM). All experiments used cells at passages 4–7. 

### 4.4. Cell Treatment and Viability

FLSs seeded in 24-well plates were starved with serum-free medium for 24 h. Cell treatments were in fresh serum-free medium containing various concentrations of the tested compounds. Where indicated, Ki16425, HA130, and heparin were added to the culture medium 30 min before cell activation with LPA, lysoPS, and PLA1A. Cell supernatants were collected after 24 h and stored at −80 °C until IL-8 measurements by ELISA. Cell viability was evaluated by PI using flow cytometry. Cells were detached using trypsin/EDTA and incubated with PI (5 µg/mL). PI negative FLSs were considered viable.

### 4.5. ELISA

Synovial fluid samples were treated with 0.5 mg/mL hyaluronidase for 5 min at room temperature before measurement. PLA1A levels in plasma and synovial fluid samples and IL-8 levels in cell culture supernatant samples were monitored in duplicate, according to the manufacturer’ protocol together with a standard curve (0.25–8 ng/mL for plasma samples: 78.1–5000 pg/mL for synovial fluid samples and 12.5–800 pg/mL for cell culture supernatant samples). Optical densities were determined using a SoftMaxPro40 plate reader at 450 nm. 

### 4.6. Flow Cytometry

For detection of PLA1A, FLSs were cultured in a medium indicated with or without serum for 24 h and cells were detached using accutase cell detachment solution. Suspended cells were blocked with 1% fatty-acid free albumin and fixed with 0.5% PFA for 20 min at room temperature in the dark. Permeabilized cells (10 µg/mL digitonin) were stained with anti-PLA1A (1 µg for 10^6^ cells) or rabbit IgG isotype control (1:5000 dilution) for 40 min on ice, followed by PE-conjugated anti-rabbit IgG secondary antibody (1:50 dilution) for 30 min on ice. For measurements of externalized PS, FLSs were stimulated with TNF for indicated times and were detached using accutase cell detachment solution. Suspended cells were stained with PI (5 µL for 10^5^ cells) and FITC Annexin V (5 µL for 10^5^ cells) for 15 min at room temperature. FITC Annexin V-positive and PI-negative identified the PS-positive viable cells.

### 4.7. Semi-Quantitative RT-PCR

Total RNA from human primary FLSs, CD4^+^ T cells, and testis tissue (provided by Dr. Belleannée of CHU de Québec) were isolated using Ribozol and reverse-transcribed into cDNA with EcoDry premix. Semi-quantitative RT-PCR was conducted using human oligo primer sequences (Table 1) and SYBR Green SuperMix. Amplification conditions were as follows: 40 cycles at 95 °C (denaturation, 15 s), 56.7 °C (annealing, 30 s), and 60 °C (extension, 30 s). 

### 4.8. Western Blot

Where indicated, FLSs were cultured with or without serum for 24 h and detached using accutase cell detachment solution. Detached cells were lysed in boiling 2× Laemmli sample buffer (125 mM Tris/HCL (pH 6.8), 17.5% (*v*/*v*) glycerol, 10% (*v*/*v*) beta-mercaptoethanol, 8% (*v*/*v*) SDS, 0.025% bromophenol blue, 5 mM sodium orthovanadate, 0.025 mg/mL aprotinin-leupeptin protease inhibitor) for 7–10 min. Equal amounts of protein samples were separated using 12% SDS-polyacrylamide gel electrophoresis and transferred to methanol activated Immobilon PVDF membranes. Membranes were blocked with 5% skim milk at room temperature for 1 h and incubated with PLA1A antibody (overnight at 4 °C) or GAPDH antibody (one hour at room temperature), followed by appropriate HRP-conjugated secondary antibodies (30 min at room temperature). The Western Lightning chemiluminescence reagent, used according to the manufacturer’s instructions, visualized the antibody-antigen complexes.

### 4.9. Statistical Analysis

Unless otherwise stated, data were from three independent experiments and the results were expressed as mean values ± SE. All statistical analyses used GraphPad Prism 8. Student *t*-test (two-tailed *p*-value) assessed the statistical significance between two experimental conditions (treated vs control). Multiple comparisons used one-way ANOVA test. A *p*-value less than 0.05 was considered statistically significant.

## 5. Conclusions

In summary, our study demonstrated for the first time that the inflammatory responses induced by PLA1A were dependent on the ATX-LPAR axis in cultured human primary FLSs. As presented in the image (Figure 7), PS exposed on the outer leaflet of FLSs can be hydrolyzed by extracellular PLA1A to produce lysoPS, a substrate of ATX. Then ATX-derived LPA stimulated the production of IL-8 in an autocrine manner through LPA1/3-associated signaling pathways. IL-8 is a neutrophil chemotactic factor that plays a role in inflammation. The high level of plasma PLA1A in EA patients suggested a possible implication of PLA1A in RA pathogenesis and qualified PLA1A as a potential biomarker and therapeutic target in chronic rheumatic autoimmune diseases. 

## Figures and Tables

**Figure 1 ijms-22-12685-f001:**
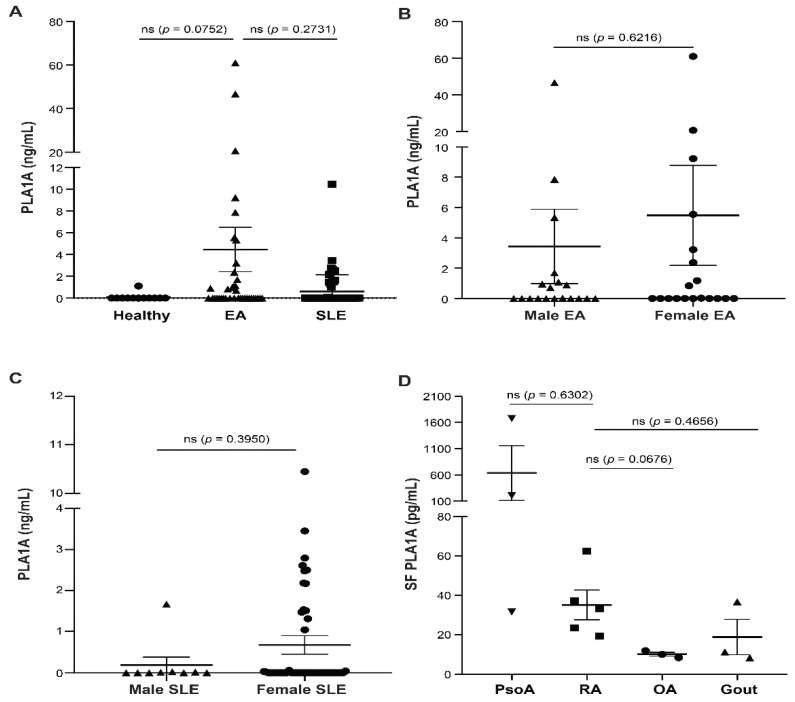
Concentrations of PLA1A in plasma and synovial fluids (SF) from arthritis patients and control donors. Concentrations of PLA1A in plasma and synovial fluids from RA patients and control groups were monitored using ELISA. (**A**) Human platelet-free plasma samples were collected from 12 healthy donors; 38 EA patients (**B**), including 19 males and 19 females); and 62 undergoing treatment SLE patients (**C**), including 9 males and 53 females. (**D**) Synovial fluid samples were collected from 3 PsoA patients, 5 RA patients, 3 OA patients, and 3 gout patients and were treated with 0.5 mg/mL hyaluronidase for 5 min at room temperature before measurement. For statistical comparative analyses, PLA1A concentrations in EA patient plasma were compared to that in healthy donors and in SLE patients. PLA1A concentrations in synovial fluid of RA patients were compared to that in PsoA, OA, and gout donors.

**Figure 2 ijms-22-12685-f002:**
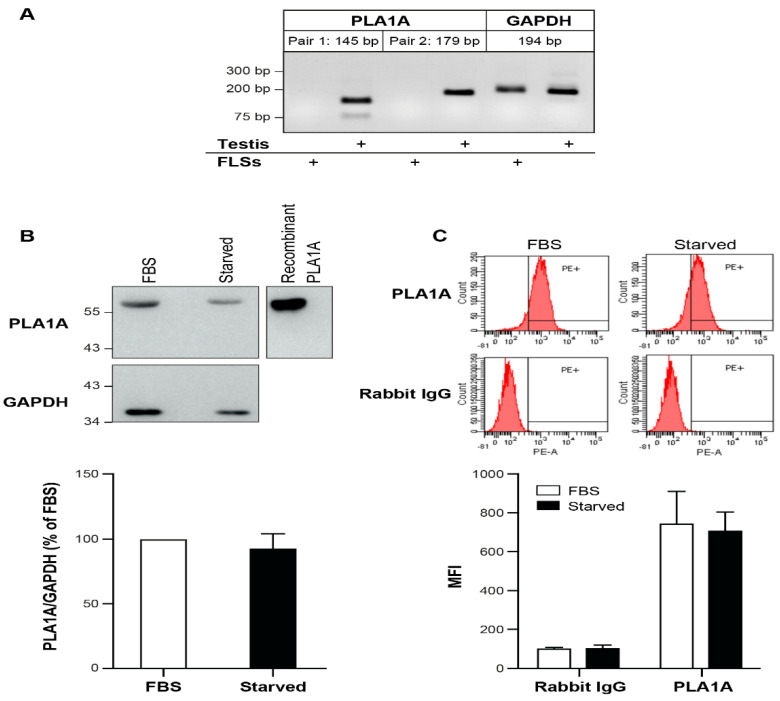
Presence of PLA1A in cultured human primary FLSs. (**A**) Total RNA from human primary FLSs from normal donors were tested for the expression of PLA1A by semi-quantitative RT-PCR. Two pairs PLA1A oligo primers were tested. GAPDH was used as a house-keeping gene. Total RNA from human testis tissue was used as a positive control. (**B**) The presence of PLA1A in human primary FLSs cultured in the absence or presence of 10% FBS were examined using western blot. GAPDH was used as an internal loading control. Recombinant PLA1A protein was used as a positive control. The blots shown are representative of five independent experiments with similar results. Amount of PLA1A was quantified densitometrically and was normalized with respect to total GAPDH. The data were the means ± SE from five experiments. (**C**) The presence of PLA1A in human primary FLSs cultured in the absence or presence of 10% FBS were examined using flow cytometry. Cells were permeabilized with digitonin, and rabbit IgG isotype was used as a control. The figures shown were representative of three independent experiments with similar results. The data were the means ± SE from three experiments. MFI, mean fluorescence intensity.

**Figure 3 ijms-22-12685-f003:**
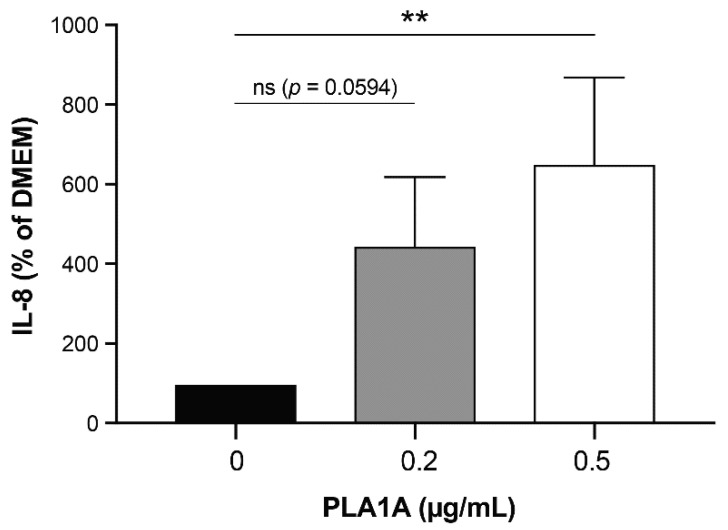
IL-8 production in human primary FLSs stimulated with PLA1A. Human primary FLSs from normal donors were stimulated with 0.2 µg/mL or 0.5 µg/mL recombinant PLA1A for 24 h. The amounts of IL-8 released in the supernatants were monitored using ELISA. IL-8 produced by FLSs cultured in serum-free DMEM was 4.03 ± 1.53 pg/mL, and for each experiment, basal IL-8 production was set at 100% for data normalization. The data shown are the means ± SE from three experiments. For statistical comparative analyses, cytokine amount in cells stimulated with PLA1A were compared to that in non-stimulated cells. ** *p* < 0.01.

**Figure 4 ijms-22-12685-f004:**
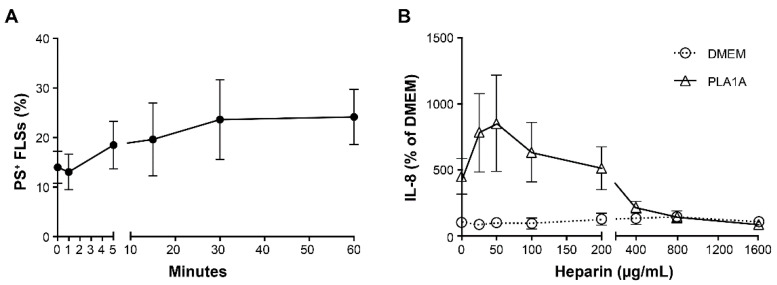
Exposure of phosphatidylserine (PS) and effects of PLA1A on activated human primary FLSs. (**A**) Human primary FLSs were activated with 100 ng/mL TNF for 1, 5, 15, 30, and 60 min. Surface exposure of PS on collected cells were stained with propidium iodide (PI) and FITC Annexin V and examined using flow cytometry. FITC Annexin V-positive and PI-negative cells were considered as PS-positive viable cells. (**B**) Human primary FLSs were treated with 0.2 µg/mL recombinant PLA1A in the absence or presence of heparin at concentrations of 25, 50, 100, 200, 400, 800, and 1600 µg/mL for 24 h. Heparin was added to cell culture 30 min prior to PLA1A. The amounts of IL-8 released in the supernatants were monitored using ELISA. IL-8 produced by FLSs cultured in serum-free DMEM was 5.37 ± 1.43 pg/mL, and for each experiment, basal IL-8 production was set at 100% for data normalization. The data were the means ± SE from three experiments.

**Figure 5 ijms-22-12685-f005:**
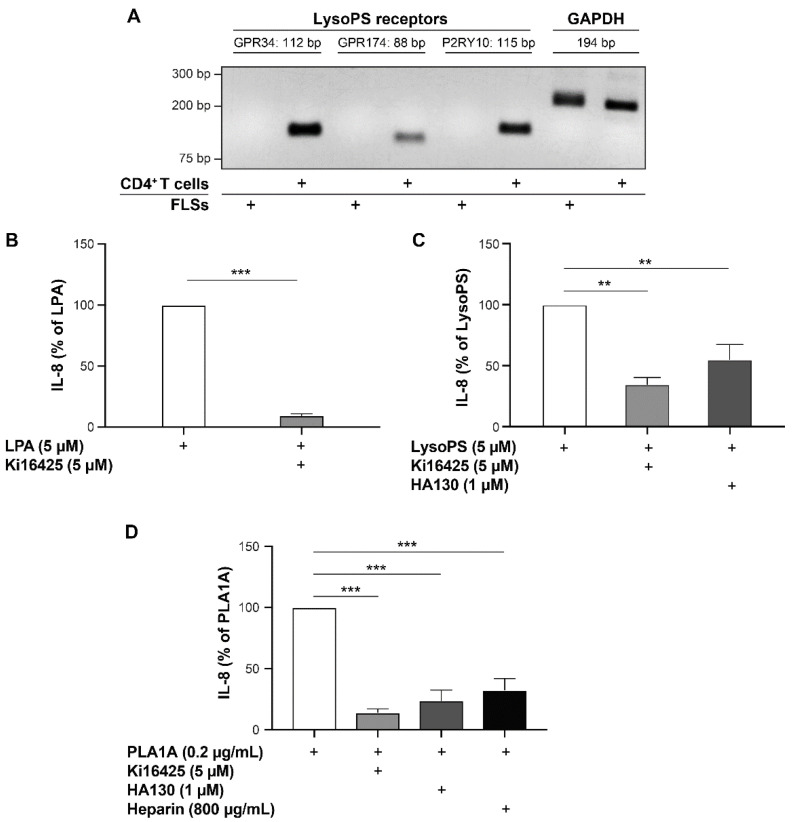
Pro-inflammatory effects of LPA, lysoPS, and PLA1A in human primary FLSs. (**A**) Total RNA from human primary FLSs from normal donors were tested for the expression of lysoPS receptors using semi-quantitative RT-PCR. For each lysoPS receptor (GRP34, GPR174, and P2RY10), three pairs oligo primers were tested. Data shown are results of one pair primer for each receptor. GAPDH was used as a house-keeping gene. Total RNA from human CD4^+^ T memory cells was used as a positive control. (**B**–**D**) Human primary FLSs from normal donors were treated with 5 µM LPA (**B**), or 5 µM lysoPS (**C**), or 0.2 µg/mL PLA1A (**D**) in the presence of 5 µM Ki16425 or 1 µM HA130 or 800 µg/mL heparin for 24 h. Ki16425, HA130, and heparin were added to cell culture 30 min prior to LPA, lysoPS, and PLA1A. The amounts of IL-8 released in the supernatants were monitored using ELISA. IL-8 produced by FLSs stimulated with LPA, lysoPS, and PLA1A was 18.17 ± 7.19, 55.65 ± 22.6, and 23.11 ± 7.44 pg/mL, respectively. For each experiment, stimulated IL-8 production without inhibitors was set at 100% for data normalization. The data are the means ± SE from three experiments. For statistical comparative analyses, cytokine amounts in cells stimulated with LPA or lysoPS or PLA1A were compared to that in cells stimulated in the presence of Ki16425 or HA130 or heparin. ** *p* < 0.01; *** *p* < 0.001.

**Figure 6 ijms-22-12685-f006:**
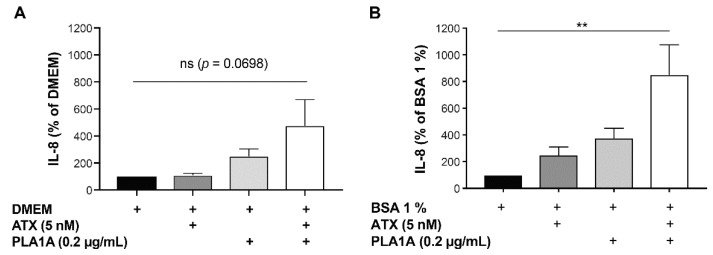
IL-8 production in human primary FLSs stimulated with exogenous PLA1A and ATX. Human primary FLSs from normal donors were stimulated with 0.2 µg/mL recombinant PLA1A in combination with 5 nM recombinant ATX for 24 h, in the absence (**A**) or presence (**B**) of 1% fatty acid-free BSA added in the culture medium. Amounts of IL-8 released in the supernatants were monitored using ELISA. IL-8 produced by FLSs cultured in serum-free DMEM supplemented with or without 1% fatty acid-free BSA was 7.08 ± 2.45 and 8.05 ± 2.04 pg/mL, respectively. For each experiment, basal IL-8 production was set at 100% for data normalization. The data are the means ± SE from five (**A**) and three (**B**) experiments, respectively. For statistical comparative analyses, cytokine amounts in cells stimulated with PLA1A and/or ATX were compared to that in non-stimulated cells. ** *p* < 0.01.

**Figure 7 ijms-22-12685-f007:**
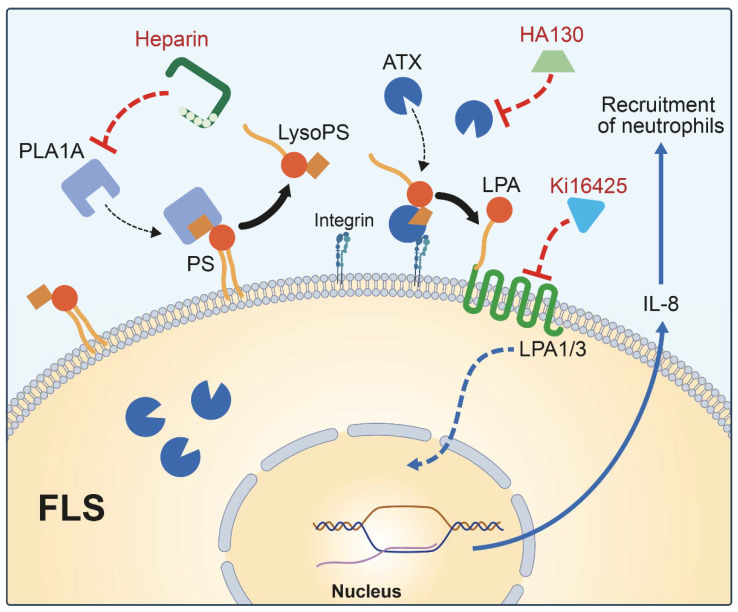
Schematic illustration of IL-8 production in FLSs stimulated with PLA1A. Activated FLSs exposed PS on outer leaflet of cell membrane and presented this specific lipid substrate to extracellular PLA1A to produce lysoPS, which was subsequently converted by additional or FLSs-secreted ATX to produce LPA. LPA stimulated IL-8 production in an autocrine manner through activating LPA1/3 associated signaling pathway. IL-8 is a neutrophil chemotactic factor that plays a role in inflammation. Added heparin competitively bound to PLA1A to prevent surface-exposed PS from hydrolyzation. ATX inhibitor HA130 and LPA1/3 antagonist Ki16425 decreased secretion of IL-8 in FLSs through inhibiting ATX-LPAR axis.

**Table 1 ijms-22-12685-t001:** List of primers used in this work.

		Sense	Antisense
GAPDH		5′-AAT CCC ATC ACC ATC TTC CAG-3′	5′-TTC ACA CCC ATG ACG AAC AT-3′
PLA1A	Pair 1	5′-GCC CAA GGA TAG GAC TGG TG-3′	5′-GCG TTG CTG CTT AGG TAT GG-3′
	Pair 2	5′-CCA TCC ACA CAG ACA CCG AC-3′	5′-TGA CAC CAC CTT GTT CCA CC-3′
GPR34	Pair 1	5′-TGT ATT TCC TGA TGT CCA GTA AT-3′	5′-GCT TTC ACT TCT GCT TGC TT-3′
	Pair 2	5′-GGG ACT GGT TGG GAA CAT AAT-3′	5′-GAT GAG TAG GAG GTC TGC AAT G-3′
	Pair 3	5′-CAG CAA CGG AAG GCA ATA ACA-3′	5′-GCT TCT CCT TTT GCG TTA TGC T-3′
GPR174	Pair 1	5′-GCC TTA TGG GTA TTT TAC GG-3′	5′-AGT CAG CAA TGG CTA GGT TT-3′
	Pair 2	5′-GGG TTT GTA ACT CCG CTT CT-3′	5′-GAT CTT GGG CCA TGG GAT ATT-3′
	Pair 3	5′-ATC AGT GTG CGA CGA TTT TGG-3′	5′-AAA GAG TAC ACA GGC AAG GCA-3′
P2RY10	Pair 1	5′-AGT CTT CGT TAT CTG CTT CAC T-3′	5′-GAT GGA AAT ACA GGG TAC TTT T-3′
	Pair 2	5′-CTC TAT GCT GGT CAT TCC CTT C-3′	5′-CCA GGG TAC ATG CTT CCT ATT C-3′
	Pair 3	5′-TAG CAG TTG TCC CGT TGT CC-3′	5′-CAT GGC GGG ATA GTT GGT CA-3′

## Data Availability

All data are included in the manuscript.

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
