# Peer review of "Phospholipase A1 Member A Activates Fibroblast-like Synoviocytes through the Autotaxin-Lysophosphatidic Acid Receptor Axis"

_ijms, 2021, doi:10.3390/ijms222312685_

Round 1

Reviewer 1 Report

The pathophysiology of PLA1A (PS-PLA1) levels in blood has been thoroughly analyzed by Yatomi and his colleagues, who included SLE, OA, and RA, making the measurement of PLA1A in clinical samples in this study less novel. However, it is worth noting that PLA1A was high in the EA. In fact, in the present study, PLA1A was significantly elevated in EA (about 50-fold) compared with normal subjects. They also showed a similar elevation of PLA1A in RA (3.5-fold). Sawada and Yatomi et al. reported that PLA1A (PS-PLA1) is not elevated in RA. This is inconsistent with the present study. The number of RA samples in the present study was quite small (n = only 3), and with this number of samples, it is not possible to conclude that PLA1A is elevated in RA.

It is interesting to note that the recombinant PLA1A protein stimulates the production of IL-8 from human primary FLSs. However, the importance of the PLA1A-LysoPS axis for IL-8 production is unclear, as the actual level of IL-8 production is not shown. The use of positive control reagents, which induce IL-8 production in human primary FLSs, can evaluate the IL-8 production by PLA1A.

The hypothesis that PLA1A produces LysoPS and that ATX converts the LysoPS to LPA has already been proposed by Aoki et al. In an in vitro system using a serum-free medium with low amounts of lysophospholipids such as LPC, it is natural that LPA is produced by ATX when PLA1A or LysoPS is added. Authors need to show whether the PLA1A-LysoPS-ATX-LPA-LPAR axis actually works under in vivo-like conditions (e.g. in the presence of serum), rather than under serum-free experimental conditions.

Author Response

We thank the reviewers for their generous comments on the manuscript. We have addressed the concerns of the reviewers and included more details to certain parts. We are now submitting a revised and substantially improved version of the manuscript.

Our detailed “point-by-point” responses to all reviewers’ comments and out rebuttals are outlined below.

Reviewer 1:

The pathophysiology of PLA1A (PS-PLA1) levels in blood has been thoroughly analyzed by Yatomi and his colleagues, who included SLE, OA, and RA, making the measurement of PLA1A in clinical samples in this study less novel.

We appreciate the comment of the reviewer and we agree that Yatomi and his colleagues have analyzed the presence of PLA1A in several autoimmune diseases, such as SLE, RA, Sjögren’s syndrome, and systemic sclerosis [8]. In their article, the authors emphasized the differences between treated and untreated SLE patients. However, in our manuscript, we emphasized the early diagnosed RA (EA) patients, who had not received any anti-rheumatic treatments. Besides, we measured plasma but not serum PLA1A, which exclude the possible bias caused by the release of intracellular PLA1A during blood coagulation.

However, it is worth noting that PLA1A was high in the EA. In fact, in the present study, PLA1A was significantly elevated in EA (about 50-fold) compared with normal subjects. They also showed a similar elevation of PLA1A in RA (3.5-fold). Sawada and Yatomi et al. reported that PLA1A (PS-PLA1) is not elevated in RA. This is inconsistent with the present study.

Sawada et al reported that serum PLA1A was not elevated in RA compared to healthy controls [8]. In their article, the authors measured the PLA1A levels in “active RA patients”, but there was no more description of the disease stage of these patients, disease duration, etc. In our manuscript, we monitored the PLA1A in EA patients (symptoms £ 12 months), who had not received any disease-modifying antirheumatic drugs (DMARDs). Furthermore, joint manifestations in patients with early disease are difficult to distinguish from other forms of inflammatory polyarthritis. We believe that the early stage of the disease and the uncontrolled inflammation could cause the increase of PLA1A level. A high PLA1A level is reported in untreated (incident cases) compared to treated SLE patients (prevalent cases) [8]. Besides, we measured the PLA1A level in the plasma but not in the serum, which excluded the release of intracellular PLA1A by blood cells during serum preparation.

The number of RA samples in the present study was quite small (n = only 3), and with this number of samples, it is not possible to conclude that PLA1A is elevated in RA.

We measured the PLA1A in synovial fluid samples from 5 RA patients, 3 osteoarthritis patients, 3 gout patients, as well as 3 psoriatic arthritis patients. In the revised manuscript, we added the group of psoriatic arthritis in Figure 1D and modified relevant sentences in Figure 1 legend, 2nd paragraph of section 2.1, and section 5.2. We agree that the present study is a small sample study. However, the improvements of biologic treatments largely replace the invasive medical treatments, such as the intra-articular injection, making it more difficult to get synovial fluid samples nowadays.

It is interesting to note that the recombinant PLA1A protein stimulates the production of IL-8 from human primary FLSs. However, the importance of the PLA1A-LysoPS axis for IL-8 production is unclear, as the actual level of IL-8 production is not shown.

The actual level of IL-8 production by untreated FLSs (group DMEM) or cells treated with 0.2 µg/ml PLA1A was indicated in the 1st paragraph of section 2.2 and Figure 3 legend. The IL-8 production by cells treated with lysoPS was indicated in the 2nd paragraph of section 2.3 and the Figure 5 legend.

The use of positive control reagents, which induce IL-8 production in human primary FLSs, can evaluate the IL-8 production by PLA1A.

We appreciate the comment of the reviewer. LPA is well known to stimulate the production of IL-8 in several cells, including the FLSs. In Figure 5 legend, we indicated the actual level of IL-8 produced by LPA stimulated FLSs. There is an error Y-axis title in Figure 5B. We are sorry about this error that is corrected in the revised manuscript.

The hypothesis that PLA1A produces LysoPS and that ATX converts the LysoPS to LPA has already been proposed by Aoki et al. In an in vitro system using a serum-free medium with low amounts of lysophospholipids such as LPC, it is natural that LPA is produced by ATX when PLA1A or LysoPS is added. Authors need to show whether the PLA1A-LysoPS-ATX-LPA-LPAR axis actually works under in vivo-like conditions (e.g. in the presence of serum), rather than under serum-free experimental conditions.

We agree that the catalytic properties of PLA1A have been reported. In the study, we focus on the involvements of PLA1A in the synovial environment. The aim was to understand the functions of PLA1A in the pathogenesis of synovial inflammation. Our data suggest that within the synovial environment, PLA1A can hydrolyze PS exposed at the surface of synovial and contribute to the autocrine ATX-LPA-LPAR axis. The use of serum is not ideal for several reasons. First, the presence of PLA1A and ATX as well in the serum cannot be avoided. Second, there are plenty of factors such as LPA and sphingosine-phosphate, which are known to induce a receptor-mediated IL-8 secretion by synoviocytes. There are also substantial IL-8 and IL-1b levels in the serum, which can also introduce a bias to the results. Third, there is a lot of LPC in the serum (and synovial fluids as well). PLA1A does not produce LPC. High serum levels of LPC would not have allowed detection of the PLA1A metabolic pathway and its potential in synovial inflammation. We don’t believe that serum represents an informative in vivo like condition. The only way to prove that PLA1A plays a role in synovial inflammation in vivo would be to develop the Pla1a KO mice and demonstrate that Pla1a KO mice are less prone to arthritis. We generated Pla1a KO mice but the experiments were delayed due to COVID-19 and the genetic background of the mice (C57Bl/6) is not appropriate for using the mouse model of collagen-induced arthritis. The phenotype of Pla1a KO mice is currently under analysis using other mouse models of arthritis. 

Reviewer 2 Report

Dear authors 

You kindly submitted a research article entitled,"Phospholipase A1 member A activates fibroblast-like synoviocytes through the autotaxin-lysophosphatidic acid receptor axis" which is interesting and clinically innovative. To investigate the possible role of PLA1A during rheumatic diseases, the authors monitored PLA1A levels in synovial fluids from patients with rheumatoid arthritis (RA) and plasma of patients with early arthritis and clinically stable systemic lupus erythematosus. They found the PLA1A level was higher in RA than those in osteoarthritis and gout, while PLA1A levels in EA and SLE patients were elevated compared to healthy donors. Further, the authors used in vitro experiments with human primary fibroblast-like synovial cells to evaluate the PLA1A-induced biological responses and  suggest PLA1A-mediated pro-inflammatory responses through the ATX-LPA receptor signaling axis. The study is generally acceptable, and only a few comments are listed as follows:

  1. Since the authors had investigated the human tissues and performed in vitro experiments, why do the authors not consider to establish an animal model with arthritis to provide more solid data?
  2. Recommend to add following two articles for precise citations of ATX-LPA axis in cancer after reference 28.  1) Lin YH, Lin YC, Chen CC. Lysophosphatidic Acid Receptor Antagonists and Cancer: The Current Trends, Clinical Implications, and Trials. Cells. 2021 Jun; 10(7):1629. 2)Wu PY, Lin YC, Huang YL, Chen WM, Chen CC, Lee H. Mechanisms of lysophosphatidic acid-mediated lymphangiogenesis in prostate cancer. Cancers. 2018 Oct; 10(11):pii:E413.

Author Response

Since the authors had investigated the human tissues and performed in vitro experiments, why do the authors not consider to establish an animal model with arthritis to provide more solid data?

We appreciate the comment of the reviewer. To study the functions of PLA1A, we generated a Pla1a KO mouse. Our current data suggest that Pla1a KO mice are less susceptible to arthritis. However, further analyses are required to study the mechanism by which PLA1A promotes inflammation. Deep analyzes of mouse phenotypes are required since the knockout of Pla1a would affect the production of lysoPS. LysoPS receptors are expressed differentially by leukocytes. In addition, to contribute to LPA synthesis, local changes in lysoPS production may directly impact the differentiation and function of immune cells.

Recommend to add following two articles for precise citations of ATX-LPA axis in cancer after reference 28.  1) Lin YH, Lin YC, Chen CC. Lysophosphatidic Acid Receptor Antagonists and Cancer: The Current Trends, Clinical Implications, and Trials. Cells. 2021 Jun; 10(7):1629. 2)Wu PY, Lin YC, Huang YL, Chen WM, Chen CC, Lee H. Mechanisms of lysophosphatidic acid-mediated lymphangiogenesis in prostate cancer. Cancers. 2018 Oct; 10(11):pii:E413.

We thank the reviewer for the suggestion. We included these two articles in the revised manuscript.

Reviewer 3 Report

The functions of PLA1A has been associated with different pathological processes in one of the most exciting fields of biomedical sciences nowadays.

The Introduction section is precise, concise and conveys the readers toward the focus of the study: the role of ATX-LPAR axis.

Even though Figure 7 (cartoon) is illustrative, I recommend more detailed discussion on the role of IL-8.

My only concern is Figure 6A. I am convinced that the lack of significance is due to the small n (n = 3). The error is high in the group (DMEM+ATX+PLA1A) (3rd column) vs DMEM alone (1st column). Please add more data. In significance appears authors must go deeply in their Discussion (possibly with changes).

Author Response

The functions of PLA1A has been associated with different pathological processes in one of the most exciting fields of biomedical sciences nowadays.

The Introduction section is precise, concise and conveys the readers toward the focus of the study: the role of ATX-LPAR axis.

Even though Figure 7 (cartoon) is illustrative, I recommend more detailed discussion on the role of IL-8.

We appreciate the comments of the reviewer. As suggested, in the 4th paragraph of section 3, we added more details to the discussion section on the roles of IL-8 in the pathogenesis of arthritis and recruitment of neutrophils (new references [41] [42] [43]). Figure 7, the legend of Figure 7, and section 4 were also modified to highlight the function of IL-8, a chemokine that recruits leukocytes to sites of inflammation.

My only concern is Figure 6A. I am convinced that the lack of significance is due to the small n (n = 3). The error is high in the group (DMEM+ATX+PLA1A) (3rd column) vs DMEM alone (1st column). Please add more data. In significance appears authors must go deeply in their Discussion (possibly with changes).

As suggested by the reviewer, we added two new experiments to Figure 6A and reached n = 5. The data areup-dated (section 2.4 and Figure 6 legend). As shown in the revised Figure 6A, with more data, the variation of group DMEM+ATX+PLA1A is decreased, but the p-value is still > 0.05. Synoviocytes are slow-growing cells. It was not possible to do more than two experiments within the ten days allocated for revision of the article.

The combination of ATX and PLA1A can increase the production of IL-8 to a certain extent in the absence of albumin. The presence of albumin increases IL-8 production possibly due to the expression of lipid phosphate phosphatases by synoviocytes, which reminds the importance of albumin as a protein that enhance the half-life of lysophospholipids. As suggested, a deeper discussion was added to the 6th paragraph of section 3. 

Round 2

Reviewer 2 Report

Dear editors 

The authors submitted a research article entitled, "Phospholipase A1 member A activates fibroblast-like synoviocytes through the autotaxin-lysophosphatidic acid receptor axis" which is interesting and clinically innovative. To investigate the possible role of PLA1A during rheumatic diseases, the authors monitored PLA1A levels in synovial fluids from patients with rheumatoid arthritis (RA) and plasma of patients with early arthritis and clinically stable systemic lupus erythematosus. They found the PLA1A level was higher in RA than those in osteoarthritis and gout, while PLA1A levels in EA and SLE patients were elevated compared to healthy donors. Further, the authors used in vitro experiments with human primary fibroblast-like synovial cells to evaluate the PLA1A-induced biological responses and  suggest PLA1A-mediated pro-inflammatory responses through the ATX-LPA receptor signaling axis.

The study is logically reasonable and generally acceptable, although the novelty is fair. The revision also explained why they didn't establish and perform animal studies.  The reviewer can accept their choice and present work.

Reviewer 3 Report

I agree with the authors' reply.